# Stable Bias:
# Evaluating Societal Representations in Diffusion Models

**Alexandra Sasha Luccioni**
Hugging Face, Canada
`sasha.luccioni@huggingface.co`

**Christopher Akiki**
Leipzig University and ScaDS.AI
`christopher.akiki@uni-leipzig.de`

**Margaret Mitchell,**
Hugging Face, USA

**Yacine Jernite,**
Hugging Face, USA

## Abstract

As machine learning-enabled Text-to-Image (TTI) systems are becoming increasingly prevalent and seeing growing adoption as commercial services, characterizing the social biases they exhibit is a necessary first step to lowering their risk of discriminatory outcomes. This evaluation, however, is made more difficult by the synthetic nature of these systems' outputs: common definitions of diversity are grounded in social categories of people living in the world, whereas the artificial depictions of fictive humans created by these systems have no inherent gender or ethnicity. To address this need, we propose a new method for exploring the social biases in TTI systems. Our approach relies on characterizing the variation in generated images triggered by enumerating gender and ethnicity markers in the prompts, and comparing it to the variation engendered by spanning different professions. This allows us to (1) identify specific bias trends, (2) provide targeted scores to directly compare models in terms of diversity and representation, and (3) jointly model interdependent social variables to support a multidimensional analysis. We leverage this method to analyze images generated by 3 popular TTI systems (Dall·E 2, Stable Diffusion v 1.4 and 2) and find that while all of their outputs show correlations with US labor demographics, they also consistently under-represent marginalized identities to different extents. We also release the datasets and low-code interactive bias exploration platforms developed for this work, as well as the necessary tools to similarly evaluate additional TTI systems.

## 1 Introduction

Diffusion-based approaches are one of the most recent Machine Learning (ML) techniques in prompted image generation, with models such as Stable Diffusion [52], Make-a-Scene [24], Imagen [53] and Dall·E 2 [50] gaining considerable popularity in a matter of months. These generative approaches are inspired by the principles of non-equilibrium thermodynamics [66] and trained to reverse the gradual addition of noise that is layered onto an image. One key difference that has led to the widespread adoption of diffusion models is that they are simpler to train than previous generations of generative models owing to their likelihood-based loss function, whose mode-covering characteristic [38] is also one of the main reasons why model outputs are so diverse compared to other classes of generative models. However, this iterative approach also makes it particularly difficult to directly access the latent space of the model, making it difficult to directly access and analyze the latent space of diffusion models.

Both the training and inference functions of diffusion models also rely on additional phases of text and image processing (e.g. safety filters, embeddings, etc.), which is why we refer to them as *Text-to-Image (TTI) Systems* as opposed to models: they are an assemblage of components and modules that all contribute to generating the final image, and it is hard to disentangle which module is responsible for downstream

37th Conference on Neural Information Processing Systems (NeurIPS 2023) Track on Datasets and Benchmarks.

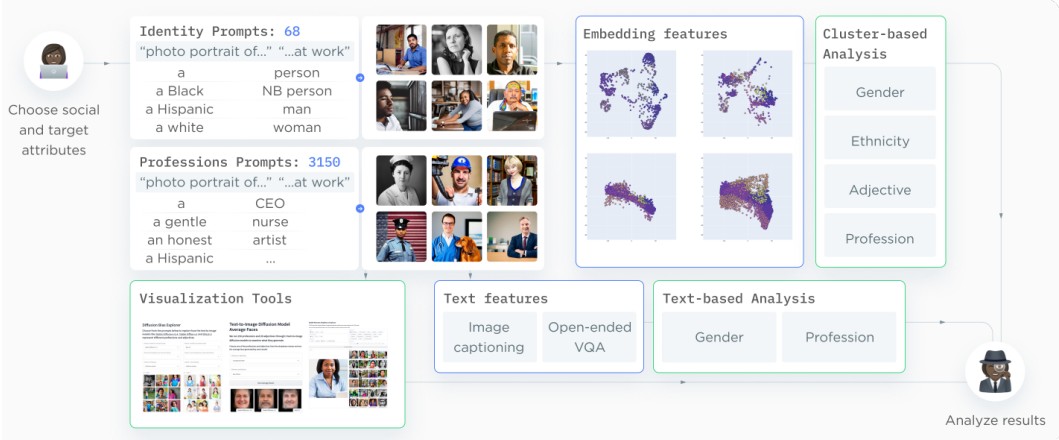

**Figure 1:** Our approach to evaluating bias in TTI systems

behavior, and what kind of biases are baked in due to models such as CLIP [29] being used to guide training. Many of these systems are also increasingly finding their way into applications ranging from generating stock imagery [35] to aiding graphic design [43] and used to generate realistic and diverse images based on user prompts. And as with any ML artifact—or, indeed, technology in general—TTI systems exhibit biases, and widely deploying them in different sectors makes them particularly prone to amplifying and perpetuating existing societal inequities. However, these biases remain sparsely documented and hard to audit, often only described in broad terms in model cards [20, 41] and in papers describing new models [53].

In the present article, we introduce a set of tools and approaches to support the auditing of the social biases embedded in TTI systems, by comparing model outputs for a targeted set of prompts (see Fig 1). We illustrate our approach by comparing images generated by Stable Diffusion v.1.4, v.2, and Dall·E 2in terms of the social biases they encode through the lens of different professions and explicit mentions of words related to gender and ethnicity. We also share tools that enable users to explore other TTI systems and topics – contributing towards lowering the barrier to entry for exploring these systems – as well as the datasets of profession generations for all of the TTI systems analyzed. We conclude with a discussion of the implications of our work and propose promising future research directions for continuing work in this space.

## 2 Background

Bias in ML is a complex concept, encompassing disparate model performance across different categories, social stereotypes propagated by models, and (mis-)representations present in datasets that can then be encoded by models and misunderstood by users. There has been extensive and insightful work analyzing the biases of ML models, both from the ML community and from many fields at the intersection of technology and society, such as social science, cognitive science, law, and policy (e.g. [54, 26, 21, 18, 58], among many others). Our project builds upon this work in ML – we endeavor to briefly describe the most relevant findings in the current section, and refer readers to further readings for more details about these topics.

### 2.1 Bias Evaluation in Multimodal ML

While it has been well-established that text models and image models encode problematic biases independently (see [13, 12, 15, 2], among many others), the rapid increase in multimodal models (e.g., text-to-image, image-to-text, text-to-audio, etc.) has outpaced the development of approaches for understanding their specific biases. It has not yet been established whether biases in each modality amplify or compound one another, a gap that our work seeks to address. For example, if a multimodal model represents the visual concepts of "people" and "White people" in a very different space than "Black people", yet represents the linguistic concept of "people" with higher positive sentiment terms than "Black people", then a compounded outcome could be model output where Black people are depicted more negatively than other identity groups.

Indeed, there is support for this concern: past work on multimodal vision and language models found that societal biases regarding race, gender and appearance propagate to downstream outputs in tasks such as image captioning (e.g., [28, 9]) and image search (e.g., [70, 42]), and has found that such biases are learned

by state-of-the-art multimodal models [67, 68]. For instance, in the case of image captioning, models are much more likely to generate content that stereotypes the people depicted, such as by associating women with shopping and men with snowboarding [70, 28], or producing lower-quality captions for images of people with darker skin [69]. Similarly, research on biases in image search has found that algorithmic curation of images in search and digital media are likely to reinforce biases, which can have a negative impact on the sense of belonging of individuals from these groups [30, 60, 40, 64]. Given the exponential popularity and widespread deployment of image generation systems, this can translate into negative impacts on the minority groups who feel under- or mis-represented by these technologies [47].

## 2.2 Text-to-Image Systems and their Biases

The recent increased popularity of TTI systems has also been accompanied by work that aims to better understand their inner workings, such as the functioning of their safety filters [51], the structure of their semantic space [14], the extent to which they generate stereotypical representations of given demographics [10, 19, 44] and memorize and regenerate specific training examples [16, 62]. Work in the bias and fairness community has also examined how models such as CLIP [49], which are used to guide the diffusion process, encode societal biases between race and cultural identity [1, 67, 68]. Some initial research has attempted to propose approaches for debiasing text and image models [8, 4, 63], as well as reducing the bias of images generated by Stable Diffusion models by exploring the latent space of the model and guiding image generation along different axes to make generations more representative [14, 55], although the generalizability of these approaches is still unclear.

One of the reasons for this is that there are many sources from which biases can originate in TTI systems; for instance, the training data used for training diffusion such as Stable Diffusion is scraped from the Web and has been shown to contain harmful and pornographic content [11, 37] as well as mislabeled and corrupted examples [59]. These datasets then undergo filters to isolate subsets that are considered, for example, 'aesthetic' or 'safe for work' [1], which are created based on the output of classification models trained or fine-tuned on other datasets [56], which can itself result in further unintended consequences. For instance, the creators of Dall·E 2 observed that attempting to filter out "explicit" content from their training data actually contributed to bias amplification, necessitating additional post-hoc bias mitigation measures [45]. Further biases can be introduced during the training process, given that many TTI systems use the CLIP model [49] to guide the training and generative process despite its biases (see above), the impact of which on image generation is still unclear. Finally, the biases of safety filters and prompt enhancement techniques used in TTI systems are poorly understood and largely undocumented [47]. For instance, keyword-based approaches have been shown to have a disparate impact on already marginalized groups [22], so using them for safety filtering at prompt level can have similar consequences.

## 3 Methodology: Auditing Social Biases in TTI Systems

In this work, motivated by the necessity to better understand and audit social dynamics in multimodal ML, we propose a new approach for quantifying the biases of TTI systems. Our approach stems from the following intuition: images generated by TTI systems may lack inherent social attributes, but they do showcase features in their depiction of human figures that viewers interpret as social markers. We cannot define these markers *a priori* due to the social nature of identity characteristics such as race or gender; they are multidimensional notions spanning a spectrum along which people choose to associate themselves [17] rather than discrete external quantities, and cannot be determined by a person's appearance. Indeed, while previous work in ML has adopted binary gender and fixed prior ethnicity categorizations for the sake of convenience, the downstream impact of these choices that are propagated from (labeled) datasets to trained models can contribute to perpetuating algorithmic harms and unfairness [7, 5, 31, 15]. We therefore endeavor to use more flexible proxy representations of the visual features in TTI systems' generated images to identify regions of the representation space that viewers may associate with social variation, and use those to audit the diversity of TTI systems in an application setting.

Our approach, shown in Figure 1, is the following: first, we define a set of *identity characteristics* for which we want to evaluate diversity and representativity – in the rest of this paper, we will consider both gender and ethnicity. Second, we generate a set of images with TTI systems by controlling the variation of markers of these social attributes in the systems' input prompts – i.e. by spanning different ways of referring to

---

[1]The criteria used to establish which images are esthetically-pleasing and safe for work are also unclear and merit further investigation.

a person's gender or ethnicity in the prompt. Third, we generate sets of images for prompts with different *social attributes* that we want to audit – here, we focus on the profession attribute and generate images for a set of 146 professions. Finally, given the multimodal nature of TTI systems, we carry out two sets of analyses: a text-based analysis that leverages Visual Question Answering (VQA) models in the text modality, and a clustering-based evaluation to characterize correlations between social attributes and identity characteristics directly in the image modality. We explain each step in more detail in the sections below.

### 3.1 Generating a Dataset of Identity Characteristics and Social Attributes

In order to generate a diverse set of prompts to evaluate the system outputs' variation across dimensions of interest, we use the pattern *"Photo portrait of a $[X]$ $[Y]$"*, [2] where $X$ and $Y$ can span the values of the identity characteristics — ethnicity and gender — and of the professional attribute that we focus our analysis on — the name of the profession. For the professional names, we rely on a list of 146 occupations taken from the U.S. Bureau of Labor Statistics (BLS), which also provides us with additional information such as the demographic characteristics and salaries of each profession that we can leverage in our analysis. For the identity characteristics, we add a suffix to make the pattern *"Photo portrait of a $[ethnicity]$ $[gender]$ at work"*– since adding the "at work" suffix makes the images more directly comparable to those generated for professions, given that these are often set in workplace settings. For gender, we use three values in this study: "man", "woman", and "non-binary person"; and as an additional option we also use the unspecified "person" to use the same pattern without specifying gender. This is still far from a complete exploration of gender variation, and in particular misses gender terms relevant to trans* experiences and gender experiences outside of the US context. For ethnicity, we are similarly grounded in the North American context, as we started with a list of ethnicities in the US census which we then expanded with several synonyms per group (the full list is available in Appendix A). Enumerating all values of the gender and ethnicity markers we defined led to a total of 68 prompts. Examples of generations for both the social attributes and target attributes sets are shown in Figure 1.

We use these prompts to generate two supporting datasets for our evaluation of three TTI systems to compare the social biases they encode: Stable Diffusion v.1.4, v.2, and Dall·E 2. We choose these three systems because they represent popular TTI systems that were state-of-the-art at the start of this project (early 2023) and allow us both to compare two versions of the same system (Stable Diffusion v.1.4. and v.2) as well as open versus closed-source systems (Stable Diffusion vs Dall·E 2). We generate an *"Identities" dataset* to ground our analysis by generating a set of images per model and combination of gender and ethnicity phrases in the prompts, for a total of 68 prompts and 2040 images. We then generate a *"Professions" dataset* to evaluate for each model, by generating a set of images for each system for each of the 146 professions in our set. In the following section, we describe our two methods for evaluating the social diversity showcased in the *"Professions" datasets* without gender or ethnicity label assignment, including a novel non-parametric method that compares its images to those in the *"Identities" dataset*.

### 3.2 Different Approaches for Analyzing Generated Images

Given the multimodal nature of TTI systems we wish to audit for social biases, we propose two methods to analyze their outputs; one method relying on text representations of the images, and one that focuses on direct comparisons in the image space; we explain these in the sections below.

**Text Features Analysis: Image-to-Text Systems** One category of image representations that we can leverage more easily to find evidence of bias is that of text-based representations; specifically, textual representations of the figures depicted in an image. We use two ML-based systems to automatically obtain such descriptions — one designed for image captioning (the ViT GPT-2 Model [46] trained on the MS-COCO dataset [33]) and one designed for Visual Question Answering, or VQA (the BLIP VQA base model [32] which was pre-trained on a set of 124M image-text pairs including MS-COCO [33] and a subset of LAION-400M [57], then fine-tuned on VQA 2.0 [25]). While the image captioning model generates multi-word captions for each of the images, the VQA system outputs a single word or a short phrase that answers the question *"What word best describes this person's appearance?"*. Both models are open-ended, which means that we do not know a priori whether the outputs will feature words directly related to the social attributes we are studying: our goal in using them is to analyze aggregate statistics of these words across the generated images, without constraining either model to labels that reflect our own biases. We also

---

[2]Before converging on this prefix, we experimented with several others, including *"Photo of"*, *"Photograph of"*, *"Portrait of"*, *"Close up of"*, but found that *"Photo portrait of"* gave the most realistic results.

recognize that both of the model contain confounding factors and biases that will be reflected in the text that they output, which is why we do not interpret the captions as a ground truth, but a feature among others.

**Visual Features Analysis: Clustering-Based Approach**  While textual descriptions of the images are much more tractable than their raw pixel representations, they also carry more limited information, especially in the case of the short VQA answers. A middle ground between the two levels of faithfulness to the information contained in an image can be achieved by leveraging dense embedding techniques that project the images into a multidimensional vector space. To that end, we leverage the same BLIP VQA system as above to also obtain image embedding; namely, the normalized average of the question token embeddings produced by the VQA encoder conditioned on the image. We chose this approach because it allowed us to focus the embedding model on the person depicted in an image - which produced an embedding structure better suited to our goal than alternatives including the CLIP image encoder [49] (see the Appendix for comparisons).

In order to make use of those embedding systems to evaluate social biases in a model's output, we need to be able to identify regions of the embedding space that correspond to visual features associated with a viewer's perception of gender or ethnicity. We do this by leveraging the *"Identities" dataset* described above. Specifically, we cluster the embeddings of the 2040 data points into 24 sets of images with a Ward linkage criterion for the dot product [65]. We can then identify trends in a TTI systems' generated images, including those in the *"Professions" dataset*, by quantifying which of these 24 regions they tend to over- or under-represent. Assigning an example to one of these regions is different from assigning it a predefined identity characteristic. Indeed, each of the region corresponds to identity prompts that span different gender and identity phrases. Additionally, delineating 24 regions in the space would not be enough to cover the 68 combinations of identity phrases showcased in the prompts. However, since the clustering was run on a space whose main source of variation came from those identity phrases, we can expect that they do encode visual features that are meaningfully associated with those – and jointly varying gender and ethnicity phrases in the prompts allows the regions to encode phenomena that are specific to their intersection. We will validate this intuition and propose specific analyses based on a TTI system outputs' cluster assignments in Section 4.2.

**Interactive Exploration**  In parallel to the approaches described in previous sub-sections, we also introduce a series of interactive tools to support story-based examination of biases in images generated by the TTI systems (which we will describe in more detail in Section 4.3). The primary goal of these tools is not to produce quantitative insights into the images, but to allow more ad-hoc in-depth explorations of them, since there are many aspects of generated images that are hard to analyze automatically (such as the presence of certain specific characteristics and visual elements in images), but which can be observed during interactive explorations, based on different angles defined by the user.

## 4   Results

The goal of our analysis is to develop ways of analyzing and comparing the biases of text-to-image systems that would allow users of these systems to shed light on these otherwise impenetrable systems. Section 4.1 compares the PROFESSION target attribute in the *"Professions" dataset* to the GENDER identity characteristic through the use of discrete textual representations of the images obtained with a captioning and a VQA system; Section 4.2 then explores the system outputs' joint variation across dimensions of GENDER and ETHNICITY following the methodology introduced in Section 3.2, and provides a detailed analysis of our 3 systems of focus as well as an overview of results for a larger set of open TTI systems. Finally, Section 4.3 describes new interactive tools that we developed based on our approach and showcases how the tools can support a qualitative analysis and storytelling around TTI system biases.

### 4.1   Gender Bias Analysis through Text Markers

As described in Section 3, we used captions and open-ended VQA to obtain textual representations of the generated images, whose discrete nature makes identifying trends related to social attribute variation more tractable. More specifically, we base our evaluation on the likelihood that a caption or VQA answer for a given profession contains gender-marked words such as 'man' and 'woman' or gender-unspecified descriptors such as 'person' or the profession name — we present these results in Table 1.

In total, 97.66% of the captions generated contained gender-marked terms, versus 45.56% of VQA appearance predictions: this is consistent given the fact that VQA mostly consists of single word predictions, whereas captions are full sentences. To put the percentage of predictions that contain

| | captions | | | VQA | | | Labor Bureau | |
|---|---|---|---|---|---|---|---|---|
| | % woman | % man | % gender markers | % woman | % man | % gender markers | % woman | % man |
| SD v.1.4 | 38.04% | 61.96% | 97.24 % | 37.77 % | 62.23% | 47.92% | | |
| SD v.2 | 33.45% | 66.55 % | 96.66% | 31.10 % | 68.90% | 44.50% | 47.03% | 52.97% |
| Dall·E 2 | 19.96% | 80.04% | 99.09% | 21.95 % | 78.05% | 44.25% | | |
| Average | 30.48% | 69.52% | 97.66% | 30.06% | 69.67% | 45.56% | | |

**Table 1:** The average percentage of mentions of 'woman', 'man', 'person' in the captions generated by a Vision Transformer Model, the BLIP VQA model and the difference between these percentages and those provided by the U.S. Bureau of Labor Statistics. N.B. these percentages are based on the number of captions/VQA appearance words containing gender markers, not the total number of data points.

gender-marked terms such as 'man' and 'woman' into perspective, we compare them with the percentages of men and women in these professions provided by the BLS [3]. We find that Dall·E 2 has the largest discrepancy compared to the BLS-provided numbers, with its captions mentioning women on average 27% less, and its VQA mentioning them 25% less, and Stable Diffusion v.1.4 having the least (approximately 9% for both captions and VQA). The professions with the biggest discrepancy between the BLS and both captions and VQA across all systems are: *clerk* (57 and 55% less), *data entry keyer* (55/53% less) and *real estate broker* (52/54% less), whereas those that have more captions that mention women are: *singer* (29/36% more), *cleaner* (20/16% more) and *dispatcher* (19/16% more).

Very few of the generated image captions mention gender-neutral terms such as 'person' (an average of less than 1% of captions for any of the systems, distributed equally across professions), and none use the 'non-binary' gender marker. Also, less than 0.5% of captions and 2% of VQA generations explicitly mention the profession in the prompt, but it is interesting to note that a single profession, *police officer*, had explicit mentions of the profession name in 80.95% of captions. In the case of VQA, several others also had a significant number of explicit mentions, including *doctor* (70.48% of VQA), *firefighter* (45.71%) and *pilot* (29.84% of VQA). We believe this to be due to the *markedness* [4] [3] of these professions and of the high proportion of gendered references to individuals (as opposed to using gender-neutral terms such as 'person') in the data used for training both TTI systems.

## 4.2 Gender and Ethnicity Distribution in the Image Space

Section 4.1 leveraged image-to-text systems to help surface a first category of biases in the output of the three TTI systems of interest. In order to further quantify the systems' diversity in terms of both GENDER and ETHNICITY without having to solve a poorly-specified and poorly-motivated identity label assignment problem, we now turn to our proposed cluster-based method to identify relevant trends in the visual features produced by these systems.

### 4.2.1 Characterizing Identity Regions in the Image Space

We apply the method described in Section 3.2 to identify significant regions in the TTI systems' output space that help summarize the variation of visual features corresponding to different combinations of gender and ethnicity phrases in the generation prompts. Specifically, we cluster images corresponding to 68 combinations of 4 phrases for gender and 17 for ethnicity into 24 regions. Thus, by quantifying the distribution of a system's generations over these regions, we can identify trends in their visual features that are correlated with the identity characteristics used to identify the regions.

Table 2 showcases this by providing summaries of 10 of the most represented regions. While the regions themselves do not correspond to specific genders or ethnicities, we can get of sense of the visual features they represent by looking at the top gender and ethnicity phrases that were featured in prompts assigned to that region in the *"Identities" dataset*. For example, we see that region 4, which accounts for over 40% of the *"Professions"* images, tends to contain images that were generated for prompts describing White men. We can further see that regions which mostly features prompts with the word **woman** make up a significantly smaller part of the *"Professions"* dataset overall (regions 15, 13, 1, and 10 together add up to 25.5%).

---

[3]The BLS only provides statistics for two gender categories on their website.

[4]Markedness is a linguistic concept that refers to the distinctiveness of a word or concept compared to others – in the case of professions, the fact that firefighters and pilots have distinctive uniforms that allow them to be easily distinguished from other professions

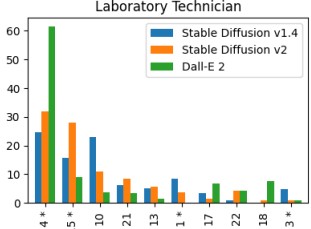

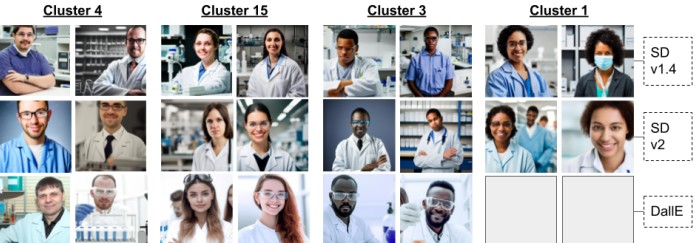

**Figure 2:** Cluster assignments of "Laboratory technician" images.

**Figure 3:** Examples of Profession images for "laboratory technician" assigned to clusters 4, 15, 1, and 3. There are no Dall·E 2 images assigned to 1.

| Region # | Share | Top gender phrases | | Top ethnicity phrases | | | |
|---|---|---|---|---|---|---|---|
| 4 | 40.1% | unspecified | man | Caucasian | White | unspecified | Latinx |
| **15** | 12.6% | **woman** | non-binary | White | Caucasian | unspecified | First Nation |
| 21 | 9.2 % | man | unspecified | unspecified | White | | |
| 18 | 8.8 % | man | unspecified | unspecified | White | Caucasian | |
| **13** | 7.5 % | **woman** | unspecified | Latinx | Hispanic | Latino | unspecified |
| 22 | 3.1 % | unspecified | man | SE Asian | Black | Indigenous | Multiracial |
| **1** | 2.7 % | **woman** | non-binary | Black | Afr.American | Multiracial | unspecified |
| **10** | 2.7 % | non-binary | **woman** | Latinx | White | Indigenous | unspecified |
| 3 | 1.9 % | man | unspecified | Black | Afr.American | Multiracial | Pacific Isl. |
| 17 | 1.3 % | non-binary | | White | unspecified | Caucasian | Hispanic |

**Table 2:** Identity clusters can be represented by the top gender and ethnicity phrases in their generation prompts. We show 10 of the main clusters above and outline those that feature the **woman** and Black gender and ethnicity phrases.

Figures 2 and 3 help us better understand how this cluster assignment functions in practice by focusing on the 630 images (210 for each system) generated for the *"Photo portrait of a laboratory technician."* prompt. Figure 2 shows the distribution over the regions separately for each of the systems, outlining a significant difference between the Dall·E 2 and Stable Diffusion outputs: with region 4 twice as prominent for the former. In order to verify our intuition about what cluster assignments mean, we visualize generations for the prompt assigned to selected regions in Figure 3, and find that the representations are coherent with the region summaries from Table 2 – Dall·E 2 is missing images for Cluster 1 because none of its generations are assigned to that cluster.

#### 4.2.2 Gender and Ethnicity Representation across Systems

The previous paragraph shows how we can use the identified regions to better understand the diversity of generations by a given system for any profession of interest. Given a prompt, cluster assignments tell us whether the model tends to generate images for this prompt that are more similar to the ones generated for a given GENDER or ETHNICITY. While this level of detail enables fine-grained analysis of a model's specific biases, more general bias trends across multiple models can be harder to apprehend at a glance.

In the following paragraphs, we propose an aggregation scheme across professions and clusters to better support such comparisons. We group professions based on the US Bureau of Labor Statistics. Rather than looking at 146 distributions for each of the professions in the list, we rank jobs based on the gender and ethnicity distributions reported by workers in the US and group them into 5 bins of 29 to 30 (quintiles). We rank and group professions by the proportion of workers who self-report as women when looking at regions of the image space that are correlated with GENDER, and by the proportion of workers who self-report as Black when examining ETHNICITY.

For regions, we create groups based on whether the corresponding gender and ethnicity phrases are prominently features in their *"Identities"* prompts (see Table 2), with **woman** and Black in the top-2 and top-4 gender and ethnicity phrases respectively. We then compare the proportion of images assigned to the grouped region per corresponding quintile to the average BLS value for this quintile. This allows us to assess not just whether a group is under- or over-represented by the models, but also whether a model attenuates, reproduces or exacerbates social biases.

Table 3 provides the results for these three analyses for both Stable Diffusion versions and for Dall·E 2. The quintile analysis makes the systems easier to compare while still retaining an important level of specificity:

| Quintiles by rank | BLS Woman | Clusters featuring "woman" phrase % | | | BLS Black | Clusters featuring "Black" phrase % | | |
|---|---|---|---|---|---|---|---|---|
| | | SD 1.4 | SD 2 | Dall·E 2 | | SD 1.4 | SD 2 | Dall·E 2 |
| Low 20% | 7.5 | 8.5 (±0.36) | 5.3 (±0.28) | 1.4 (±0.17) | 4.7 | 6.6 (±0.60) | 4.8 (±0.56) | 3.9 (±0.48) |
| 20 to 40 | 26.5 | 15.2 (±0.46) | 14.9 (±0.46) | 2.8 (±0.20) | 7.1 | 8.2 (±0.55) | 4.9 (±0.54) | 4.3 (±0.32) |
| 40 to 60 | 47.1 | 32.2 (±0.57) | 23.1 (±0.53) | 6.0 (±0.28) | 10.5 | 7.3 (±0.61) | 8.1 (±0.63) | 3.4 (±0.53) |
| 60 to 80 | 68.4 | 54.3 (±0.60) | 46.3 (±0.63) | 21.6 (±0.57) | 14.4 | 11.2 (±0.64) | 11.4 (±0.64) | 5.3 (±0.54) |
| Top 20% | 86.8 | 83.0 (±0.45) | 78.3 (±0.49) | 56.3 (±0.62) | 22.1 | 16.8 (±0.66) | 16.1 (±0.57) | 4.5 (±0.46) |

**Table 3:** Comparing cluster assignments to Bureau of Labor Statistics (BLS). Each line corresponds to one fifth of the professions grouped by BLS-reported percentage of women (left), and Black workers (middle). 95% confidence intervals per a bootstrap estimator are provided.

for example, if a system under-represents women in its outputs, this lets us know whether it is reproducing or exacerbating societal biases. Taking the example of Stable Diffusion v1.4, we can see that while the system seems to match the US distribution for the least diverse professions (whether the workforce is mostly identified as men or women), regions that feature **woman** are under-represented across the more balanced professions. We can also identify broad trends across systems: while all under-represented regions of the Space associated with the **woman** and Black phrases, the phenomenon is least pronounced for Stable Diffusion v1.4, and most pronounced for Dall·E 2.

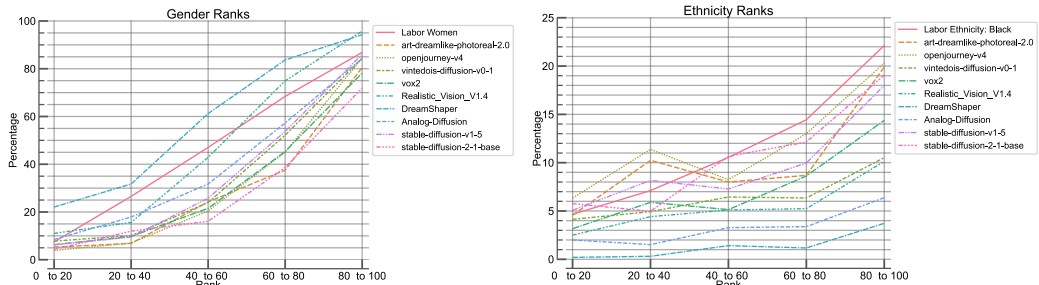

**Figure 4:** Comparing 11 TTI systems' representations of the regions associated with the **woman** and Black phrases. The format of results is the same as in Table 3 above, presented as line plots here to more easily contrast multiple models. On the "rank" x-axis, each tick corresponds to a quintile of professions grouped by the BLS-reported proportions of women (left), and Black workers (right) in those professions. The "percentage" y-axis corresponds to the percentage of image generations for those professions assigned to the clusters selected as described in Table 2 and in the text.

**Evaluating additional systems**  While we initially focused our evaluation on three TTI systems with the highest visibility at the start of this work, our method is easily applicable to any TTI system that can generate an image given a prompt – we release the necessary code for this alongside our paper. We further benchmarked an additional 11 systems selected from the most downloaded text-to-image models on the Hugging Face Hub at the time of writing. We summarize results for the same gender and ethnicity quintiles as above in Figure 4. We see that even though these systems all share Stable Diffusion models as their pre-trained initialization, the diversity of their outputs across both dimensions does seem to depend on the specific fine-tuning and adaptation process, sometimes independently from each other. These results should provide a starting point for further investigation into these systems' specific datasets and help guide specific use cases.

### 4.3  Interactive Tools for Interactive Exploration

As part of our own analysis of images generated by different TTI systems, we have created tools to help us delve into the images in more detail and identify relevant patterns to guide our analyses. We present the three of the tools that we have created, and the observations that they have allowed us to make, below:

**Diffusion Bias Explorer**  One of the first tools that we created in the scope of this project was the Diffusion Bias Explorer (See Fig. 5 (a)), which enabled users to compare what the same set of prompts – based on the list of professions described in Section 3 – resulted in when fed through the 3 TTI systems. It allowed us to uncover initial patterns – including the homogeneity of certain professions (such as CEO) or the differences between versions of Stable Diffusion as well as Dall·E 2. In fact, we used the Diffusion

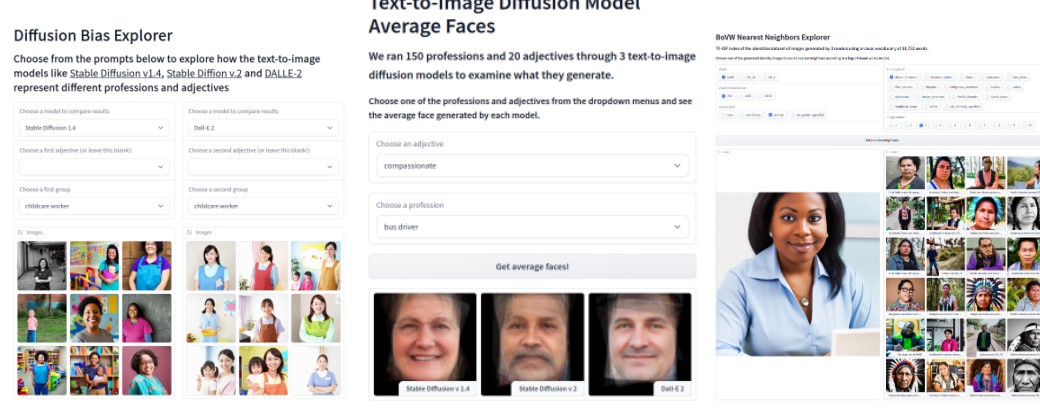

|  (a) Diffusion Bias Explorer | (b) Average Face Comparison Tool | (c) k-NN Explorer |

**Figure 5:** The 3 interactive tools created as part of our analysis: Diffusion Bias Explorer, Average Faces Comparison, and the k-NN Explorer.

Bias Explorer all throughout the project in order to visually verify tendencies discovered using our analyses, since it allowed us to quickly view images for any given profession+TTI system combination.

**Average Face Comparison Tool** Another tool that we created was the Average Face Comparison Tool (see Fig. 5 (b)), which leverages the `Facer` Python package to carry out face detection and alignment based on facial features and averaging across professions. This helped us further see more high-level patterns in terms of the visual aspects of the images generated by the different TTI systems while avoiding facial recognition and classification techniques that would prescribe gender or ethnicity. Also, the blurriness of the average images gave us signal about how homogeneous and heterogeneous certain professions were.

**Nearest Neighbors Explorers: BoVW and Colorfulness** To enable a more structured exploration of the generated images we also developed two nearest-neighbor lookup tools. Users can choose a specific image as a starting point—for example, a photo of a Black woman generated by a specific TTI system—and explore that photo's neighborhood either by colorfulness (as defined by [27]), or by a bag-of-visual-words [61] TF-IDF index. To build this index, we used a bag-of-visual-words model to obtain image embeddings that do not depend on an external pre-training dataset. We then extracted each image's SIFT features [36] and used k-means [34] to quantize them into a codebook representing a visual vocabulary and computed TF-IDF sparse representations and indexed the images using a graph-based approach [23]. These search tools enable a structured traversal of the dataset and thereby facilitate a qualitative exploration of the images it contains, either in terms of color or structural similarity. This is especially useful in detecting stereotypical content, such as the abundance of stereotypical Native American headdresses or professions that have a predominance of given colors (like firefighters in red suits and nurses in blue scrubs).

We provide code and detailed instructions about how to duplicate and edit these tools to work with other sets of prompts and images, to facilitate their dissemination and customization in the community.

## 5  Limitations and Future Work

Despite our best efforts, our research presents several limitations that we are cognizant of. First, the models that we used for generating captions and VQA answers both have their own biases (many of which we describe in Section 2.1), which we are unable to control for in our analyses. We aimed to compensate for these by leveraging multiple models and comparing their outputs, as well as by using less symbolic models such as BoVW. Second, while the open nature of the Stable Diffusion models makes us reasonably certain that we did not miss any major confounding factors in our comparative analysis, the same is not true of Dall·E 2. Given that it is only available via an API, we were unable to control for any kind of prompt injection or filtering [5], or indeed whether we were at all prompting the same model on different days. We were therefore only able to compare the output of all three models on the assumption that the images they generate correspond to the model outputs based on the input prompts. Third, our analyses are limited to a given

---

[5]A blog post from the creators of Dall·E 2in July 2022 stated that they added a technique to improve the diversity of its representations [48], with many speculating that it is based on prompt injection [59, 47].

set of social attributes in terms of gender and ethnicity, which, as we describe in Section 3, are attributes that are inherently fluid, multidimensional and non-discretizable. Finally, we recognize that none of the authors of this paper have primary academic backgrounds in scientific disciplines relevant to the social science dimensions of gender and ethnicity, and we do not have first-hand experiences for many of the identity characteristics that we refer to. We also recognized our bias towards discrimination axes that are relevant to the Western world, and the importance of extending this analysis to different cultures and different contexts.

We consider our work to be a first step in exploring societal biases in text-to-image models, with much follow up work needed to make this work more complete and nuanced. An important part of this work would be to keep exploring different dimensions and aspects of social bias, including age and visual markers of religion, as well as other target attributes that are tied to stereotypes and power dynamics. We hope that future work will continue to carefully consider the complex, interconnected nature of many types of biases and the fact that many attributes cannot be inferred visually from generated images. Finally, we believe that there is much potential work to be done in further developing interactive tools such as those we created to support qualitative analysis and storytelling around the model biases, as well as empowering stakeholders and communities with less technical expertise to engage with and probe TTI systems and other ML artifacts.

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
