# OpenReview forum: "Stable Bias: Evaluating Societal Representations in Diffusion Models"
_NeurIPS.cc/2023/Track/Datasets_and_Benchmarks — NeurIPS 2023 Datasets and Benchmarks Spotlight_

### Official Review · Reviewer_MKkt · 2023-07-17
**Neat new evaluation method for an important problem, but could interrogate limitations more thoroughly**

**Rating:** 8
**Confidence:** 5

**Strengths:**

- **[Major] Clustering analysis in particular is novel and a neat new solution to the labeling problem.** Alternative approaches usually rely on some form of messy aggregation or classification, but this approach allows relative identity comparison using variation introduced by the model itself (evaluating the model relative to its own representations of identity — really neat). The text markers approach is also novel and a useful contribution, though a bit messier (see suggestions below).
- ****[Major] The interactive tools are really useful and well constructed.**** I wish more evaluation work had interactive visuals! I used them for a bit — the tools seem very practical for red-teaming work and further applied research.
- ****[Major] Results are notable and socially significant****, if not completely novel (see discussion of related work). Methods like these have clear and pressing practical applications.
- **[Minor] Figures are excellent, especially Figure 1.** Very helpful for summarizing the methods at a high-level.

**Additional Feedback:**

N/A

**Clarity:**

The paper is very well written and easy to follow. The description of the clustering method was difficult to understand (but likely also difficult to explain), but I followed after a couple re-reads.

- **[Major] 4.2.2 is confusing.** The first paragraph was really difficult to follow, and the Fig. 4 labels/caption could be more verbose. Rank of what? Percentage of what? Why doesn’t this figure include DALL-E? The caption doesn’t help much either. It might also help to include a reference line (y=x?) where the TTI clustering perfectly matches real-world statistics, if that makes any sense.
- **[Minor] Perhaps refer to the “identities dataset” and “professions dataset” as “output” or “generated” datasets**, to make it immediately clear that they are produced uniquely by each TTI model and are not some sort of external benchmark.

**Correctness:**

As far as I can tell, the evaluation methods seem appropriate, given prior work and the methodological challenges, unless the validation experiments suggested above prove otherwise (e.g. if the captioning / VQA models produce gender markers that are completely orthogonal to self-identified gender, which seems unlikely).

**Documentation:**

The authors release a set of codebooks and datasets on Huggingface. I didn’t try all the tools but the ones I did try worked as expected out of the box and could be used to replicate or extend the results in the paper.

**Limitations:**

See notes above — the authors do mention limitations, but could do a better job contextualizing the results.

**Opportunities For Improvement:**

Take my extensive list of suggestions as a sign of my appreciation of the paper, not as a contradiction of my positive rating. Addressing the issues below would bump my rating to a 9 or 10 and make this paper all the more useful to researchers and practitioners.

- **[Major] Interrogate more deeply the entanglement of bias in the target TTI model and bias in the captioner / VQA system.** The most serious limitation of this approach — which the authors acknowledge (although only once, and not until the end of the paper) is that the results are due not only to identity/profession associations in images generated by the TTI model, but also identity/profession associations in the captioner or the VQA system used to label those generated images. As the authors mention, a central difficulty with TTI evaluation (and evaluation of text/image generation more broadly) is that the generated images do not correspond to real individuals who can self-identify. I worry that some of the results could be due to bias in the labelling model (esp. if trained on MS-COCO — see e.g. Angelina Wang’s work on biases in MS-COCO and image captioning). Is the TTI model more likely to produce an image of a female-presenting cleaner, or is the captioning model more likely to describe an image of cleaning supplies with female gender-marked language (since we know this is possible from work like Angelina’s)? How much of the results are due to one vs. the other? I don’t think this issue completely breaks the methods — for the clustering method in particular, Fig. 3 offers some clear visual encouragement that the VQA encodings are appropriate for capturing identity variations. But I think the current draft undersells and elides this limitation, so I have a couple suggestions:
    - **Systematically validate the labelling model.** One or two simple validation experiments could help convince the reader that the results are not simply due to (known) biases in the labelling models. I can think of two ways to do this: one is to simply cross-check the labeling from this approach with human evaluation (similar to Cho et al.). Or, run the analysis on a set of ****real**** faces with self-identified gender/ethnicity. (For example, Bianchi et al. use CLIP to densely embed a set of *self-identified* faces from the Chicago Face dataset. Then they compare the CLIP representations of TTI outputs to the “archetypal”, self-identified male/female face.) How do your methods behave compared to human evaluators? How well do the gender markers correlate with self-identified faces? Based on Fig. 2, I expect they will correlate quite well, but systematic results should be systematically validated.
    - **Compare and contrast with other possible methods.** To help the reader understand the authors’ choices, (briefly) compare and contrast with other possible approaches to this problem. Addressing these questions (even briefly) could better surface benefits/drawbacks of the authors’ approach.
        - The first bias evaluation of image generation that I know of (Steed & Caliskan, “Image Representations…”, FAccT 2021) follows the tradition of intrinsic bias evaluation in NLP by comparing directly the internal representations of the generating TTI model: could one apply the clustering method directly to the TTI model’s internal representation, rather than VQA encodings of the generated images? Why choose to involve a second model at all? (Perhaps it is a necessity if OpenAI/Stability doesn’t make these representations available.)
        - Other work opts for using a classifier to directly label the generated images — e.g. Cho et al. use CLIP to classify images into man/woman binary, and Bianchi et al. label man/woman based on cosine distance w/ CLIP representations. Perhaps the authors can elaborate on the benefit of leaving the labeling open-ended, compared to these approaches? And a follow-up question: it seems like even though this approach to labeling is open-ended, the analysis still involves one-hot encoding categories of interest based on gender marked text — so what’s the advantage of this two-step approach compared to a direct identity classifier? Is it any better at expressing nuances in identity? (Seems dubious, given that SOTA captioning/VQA models probably have their own gender biases.)
    - **Avoid framing the labeling process as objective — don’t elide the subjectivity of the labelling model.** The authors should avoid language that suggests the captioning/VQA results are objective, “truthful”, or even generalizable. For example, L222-224 makes it sound like the captions somehow belong inherently to DALL-E: “We find that DALL-E 2 has the largest discrepancy… with *its* captions mentioning women on average 27% less, and *its* VQA mentioning them 25% less…” (emphasis added). But both the captions and the VQA are from an external model, with its own biases — they are an interaction between DALL-E and a second model, not a property of DALL-E alone. After all, some of the results could easily be attributed to the labelling model instead — for example, the fact that the captioning model tends to rely heavily on gender binaries (L230: less than 1% of captions include gender-neutral terms) might say more about the captioning model than about the generated images (what happens if I caption images of non-binary or trans people?). The authors should also consider (briefly) mentioning the potentially confounding role of the labeling models as soon as they are first mentioned (potentially alongside some validation results).
- ******************************[Minor] Why 24 regions?****************************** How did you pick this number? How do your results change with more/fewer regions?
- **[Minor] Include standard errors / sample sizes / confidence intervals in Table 1.** Something to help the reader judge whether the differences between cells are statistically significant — whatever measure is appropriate. E.g. does DALL-E produce images that get more gendered captions, or is 99.09% about the same as 96.66% at these sample sizes?
- **[Minor] Find a way to include “non-binary” results, at least in Tables 1 & 3.** It’s great that non-binary is supported as an evaluation option (for clustering, if not for the text markers), so include it in the analysis (or say why it isn’t included, at least).

**Relation To Prior Work:**

The question of occupational bias in TTI in general is the subject of a few papers already, particularly Bianchi et al. (2023), Cho et al. (2022), and even the DALL-E system card. I still think the methods in this paper are novel and interesting, but the paper could be stronger if the authors reference these articles of prior work to substantiate the parts of their methods that are less novel (e.g. the prompting strategy) and highlight the parts of their methods that are particularly novel (e.g. more professions, the interesting clustering approach, the visualizations). Otherwise the reader might overlook the especially useful additions this paper makes.

- Bianchi et al., “Easily Accessible Text-to-Image Generation Amplifies Demographic Stereotypes at Large Scale”, FAccT ‘23. (arXiv was up before NeurIPS deadline, though published in FAccT after.) Similar prompting approach, similar comparison to BLS labor statistics, but they use a different analysis technique, different target models, fewer professions. How do these results differ? It looks like they are qualitatively the same — gender stereotypes exaggerating real-world distributions — but possibly more nuanced & intersectional.
- May also wish to mention [Naik & Nushi](https://arxiv.org/pdf/2304.06034.pdf), “Social Biases through the Text-to-Image Generation Lens”, arXiv 2023, though this one has not been peer reviewed — also covers occupational biases.

**Summary And Contributions:**

The authors propose a novel clustering and caption/VQA method for evaluating text-to-image (TTI) systems, finding that prominent TTI systems exacerbate real-world representational harms against women and black people, particularly black women, in images of occupations. The authors also provide new public, interactive tools for exploring stereotypes and biases in TTI systems.

---

> ### Author Response · Authors · 2023-08-19
>
> We would like to express our gratefulness to Reviewer MKkt for their insightful comments and suggestions and address the points raised below:
>
> **[Validating captions]**:
>
> - We did, in fact, manually cross-validate the captions that were generated by the model and did a series of analyses on top terms used for each profession and identity group. We ran early experiments where we looked at the professions inferred by the VQA model for different identity generation prompts - however, it quickly became clear that properly evaluating these biases to gain the necessary insights would have exceeded the planned scope of this paper and have therefore decided to explore that further in future work.
> - We also added a note about the confounding role of the labeling models as soon as they are first mentioned in Section 3.4.
>
> **[Clustering the TTI model’s internal representation]]**:
>
> The goal of our approach is to be applicable to both closed- and open-source models, which means that we can’t rely on any internal representations, which aren’t available for closed-source models e.g. Dall-E 2.
>
> **[Open-endedness of our approach]**:
> Given the impossibility of classifying individuals into a predefined set of identity characteristics (e.g. binary gender or a set number of race categories), we find that an open-ended approach is fitting. In our particular analysis, we chose a number of categories of interest based on the context of our study (profession + identity characteristics), but these categories can be replaced by any number of others, depending on context and interests, without utilizing an identity classifier trained on images of real people.
>
> **[Clustering approach]**:
> We actually tried several cluster numbers (as can be seen in the [Cluster explorer tool](https://huggingface.co/spaces/tti-bias/cluster-explorer) that we created), but found that 24 was the optimal number of clusters in terms of distinctiveness and interpretability of the analysis as it provided the best trade-off between granularity and interpretability for our full list of social variables (3 marked gender phrases x 16 marked ethnicity phrases), and separated social variables without separating generation models as much. Appendix B provides some evaluations for the 12-cluster and 48-cluster settings, and shows the findings to be fairly stable to the clustering size. We will discuss this choice in more detail in the final version.
>
> **[Related Work]**:
>
> Thank you for the suggestion; we will use the extra page to expand the related work section and critically engage with the suggested works.
>
> **[Clarity]**:
>
> **[MAJOR "4.2.2 is confusing"]**
> We will restructure this paragraph and extend the caption to make the following points clearer:
> - we'll align the axis labels in Figure 4 with the column names in Table 3 since they represent the same quantities. On the "rank" x-axis, each tick corresponds to a quintile of professions grouped by the BLS-reported proportions of women (left), and Black workers (right) in those professions. The "percentage" y-axis corresponds to the percentage of image generations for those professions assigned to the clusters selected as described in Table 2 and in the text.
> - We ran two sets of evaluations. SDv1.4 and 2 and DallE were first evaluated with a large number of image generations per profession, and the results are presented in Table 3. In order to further validate the method, we ran the same evaluation on an additional set of 9 models, with fewer generations (to limit the overall cost and time requirement) - these include other versions of Stable Diffusion (1.5 and 2.1), but not 1.4 or 2.0 or DallE. We did include the y=x reference line as "Labow Women" and "Labor Black" respectively.
> - We decided to present the results separately to have a stronger focus on the main result (Table 3), but we do see how the different formats make the presentation confusing and will address this further in the text.
>
> **[VQA bias]**
> - We ran early experiments where we look at the professions predicted by the VQA model for different identity generation prompts - however, it quickly became clear that properly evaluating these biases to gain the necessary insights would have constituted its own separate research project, which we would love to explore further in future work.
> - We will expand on that in the final version while also improving the language in the paper to resolve such ambiguities as the one raised by the reviewer regarding lines 222-224.
>
> **[Standard errors / Sample Sizes / CI in Table 1]**:
> Thank you for the suggestion! We used a bootstrap estimator to obtain a 99% confidence interval for the entropy comparisons in the Appendix (Table 6), we'll add similar confidence intervals for the results in the main text.
>
> **[Tables 1 and 3 -- non-binary results]**:
> We omitted the "non-binary" results from Tables 1 and 3 as there is no corresponding category in the BLS labor statistics, we will mention this reasoning in the text.

---

### Official Review · Reviewer_M7wi · 2023-07-21
**Necessary work**

**Rating:** 8
**Confidence:** 3
**Correctness:** The evaluation methods performed in t…

**Strengths:**

This is a very much needed work and I am glad that the authors took the time to perform this analysis and share their methodology.
The work is significant not only for people working in the same areas, but also for people from other disciplines that want to get an understanding of the possible societal biases in text-to-image models. The research seems to be very solid and the motivations are well-explained. The ethical and social implications are, in my opinion, positive.

**Additional Feedback:**

I am happy to participate in the open discuss of the paper. However, I will have no internet access between August 2nd to August 15th. I will be available to get back to any question or concern after August 16th.

**Clarity:**

The paper is well-written and clear.
Line 139 has a * but it's not clear where it leads.
Line 93 has wrong quotes.

**Documentation:**

The URL provided by the authors has an easy interface that allows the readers to understand the work more deeply.

**Ethics:**

I believe that the checklist is missing. Please, provide it in the revision.

**Limitations:**

The authors provide a pretty good overview of the limitations of their work.

In lines 341-343 the authors point out that none of them has a background in social sciences. As I previously stated, this work is very much needed and interesting. However, I believe that interdisciplinary work as this one would extremely benefit from the collaboration with people from the humanities and in social sciences. While I do imagine that establishing such collaborations can sometimes be hard, I believe that it should be an effort of the people working on this topic to encourage others to a more open scientific agenda.

In addition, I believe that the authors do not mention that this work has a strong (yet natural) bias towards the discrimination axes typical of the western world. Maybe we should mention that this is the frame in which such a work is conceived and that future work (or parallel work) might explore the possibility of extending this to different cultures and different contexts.

**Opportunities For Improvement:**

I believe that the work can be and should be published as is. I don't have major suggestions for improvement.

**Relation To Prior Work:**

The authors provide great motivation for their work and a good overview of the existing efforts. In particular, they divide the "Background" section in different sub-section, allowing the reader to understand the topics more easily.

**Summary And Contributions:**

The authors propose a methodology to evaluate existing biases in text-to-image models, as well as an interactive platform to reproduce the analysis. Their paper suggests that these models unfortunately hide several levels of biases.

---

> ### Author Response · Authors · 2023-08-10
>
> We truly appreciate Reviewer M7wi’s review and comments regarding our paper.
>
> We have added the minor points raised by the reviewer, including our bias towards Western perception of identity groups and professions (in the Limitations) and the small mistakes on lines 93 and 139. We have also added a paper checklist to our submission.

---

> > ### Comment · Reviewer_M7wi · 2023-08-26
> > **Thank you**
> >
> > Dear authors, thank you. I confirm my initial mark.

---

### Official Review · Reviewer_JWck · 2023-07-22
**Interesting annotator-free method of evaluation**

**Rating:** 5
**Confidence:** 3

**Strengths:**

The proposed method described the paper to analyze TTI models is accessible and scalable - it does not rely on human annotation, lowering the monetary barrier to fairness evaluation.  VQA and Image Captioning have previously been used to measure the quality of TTI models. This work leverages them specifically for fairness, and focusing on demographic representation. An easy, annotation free method of evaluating TTI models will encourage researchers to do more analysis of their own models.   While the concept of under and over representation of gender and racial groups in image generation is not new, the proposed measurement and analysis is novel.



**Additional Feedback:**

N/A

**Clarity:**

however, there are some key details that are confusing.

The paper claims that they better represent the spectrum of values across binary and ethnic categorizations through clustering, but the clustering is summarized by the same discrete quantities. Can the authors clarify the difference between the clustering.


The prompt engineering used to create the two sets is a bit confusing. What prompts are used to generate the professions dataset? Figure 1 and the Appendix mention adjectives, but they are not referenced in the text. In figure 1: "photo portrait of a Hispanic CEO at work" appears like it would be one of the generation texts, how did the authors ensure there was enough representation of each demographic pairing per profession. If the adjectives were used, were they further balanced based on the "genderedness" of the adjectives?

Can the authors please clarify that my understanding is correct: the Processions dataset is used in 4.1 and the identities dataset is only used to generate the embedding clusters in 4.2?

Appendix F overflows off the page on the left, and data is cut off.

**Correctness:**

Based on Table 9 in the Appendix, it appears that many of the clusters are dominated heavily by one generative model, >80%. Given that it appears that the clusters split by model, what is the rational for clustering them all together? Would it not make more sense to generate a fixed number of clusters per model in its own embedding space based on the Identities images from the same dataset, and cluster the professions images per model based on that?

**Documentation:**

There are a number of links to the data given. It is unclear which ones are the dataset, and how they should be used. There is not a datacard for the data generation methodology intended and out of scope use cases for documentation. It is unclear what that authors intend the generated data to be used for, as it was generated for a specific model, and analysis of new TTI models would require generating new images.

**Ethics:**

Since the authors are generating images of people, and labelling them with machine generated demographic labels, I advise the authors to create a datacard summarizing their data and making clear that the images are synthetic, and that the labels are model generated and potentially noisy - to prevent any unintended uses (ie not realizing the data is synthetic).

**Limitations:**

The authors address the limitations of using captioning and VQA models for generating labels, and the biases that these models can have. Additionally, they address the limited control that they ave over the DallE2 API. Many datasets with person-related fairness labels are released for evaluation only [1] as there is potential for misuse. It would be nice if they authors included a discussion on any potential for misuse of their dataset, or justification as to the generative nature of the images prevents misuse.


[1] Schumann, Candice, et al. "A step toward more inclusive people annotations for fairness." Proceedings of the 2021 AAAI/ACM Conference on AI, Ethics, and Society. 2021.

**Opportunities For Improvement:**

Why was 24 chosen from the number of clusters? Did the authors experiment with smaller and larger values? What were the tradeoffs of their findings based on cluster size?

Based on Table 9 in the Appendix, it appears that many of the clusters are dominated heavily by one generative model >80%.


The authors note that image captioning and VQA models have been shown to have biases. The authors should include some sort of estimate as to the model bias to better understand the confidence in their analysis. The authors could do this by both analyzing the accuracy of the captioning model on an existing fairness dataset labelled with these protected attributes as a baseline. Depending on the dataset used, this would not account for the bias in context - ie over-captioning men as doctors, and so I would also suggest the authors manually check the captioning demographic accuracy on some small portion of the dataset themselves. The authors could also consider using [1] to de-bias the existing captioning model.


[1] Hirota, Yusuke, Yuta Nakashima, and Noa Garcia. "Model-Agnostic Gender Debiased Image Captioning." Proceedings of the IEEE/CVF Conference on Computer Vision and Pattern Recognition. 2023.

**Relation To Prior Work:**

Yes.

**Summary And Contributions:**

The authors explore societal, gender, and racial biases of generative models. They explore the frequency that different demographic groups are represented in TTI images, and compare it to a baseline, the bureau of labor statistics. The use an image captioning and VQA model to analyze gender representation in image generation of professions. They compare embeddings of the generated images clusters of embeddings of generated images seeded with gender/race prompts to further analyze the representation.

The contributions are:
1) A method for analyzing gender coverage of generated images using VQA/Image captioning models
2) A method for analyzing demographic coverage across multiple attributes by clustering.
3) The dataset of generated images they used for the analysis, and visualization tools.

---

> ### Author Response · Authors · 2023-08-19
>
> We thank the reviewer for their insightful feedback and provide more details regarding their questions and comments below:
>
> **[Clustering approach]**:
>
> > Why was 24 chosen from the number of clusters? Did the authors experiment with smaller and larger values? What were the tradeoffs of their findings based on cluster size?
>
> We did experiment with different cluster sizes, Appendix B provides some evaluations for the 12-cluster and 48-cluster settings, and shows the findings to be fairly stable to the clustering size. We found that 24 provided the best trade-off between granularity and interpretability for our full list of social variables (3 marked gender phrases x 16 marked ethnicity phrases), and separated social variables without separating generation models as much. We will discuss this choice in more detail in the final version.
>
> > Based on Table 9 in the Appendix, it appears that many of the clusters are dominated heavily by one generative model >80%.
>
> There is only one such cluster in the 12-cluster setting, and they account for less than 15% of clusters in the 24- and 48-cluster setting. Table 9 shows the 48-cluster setting, where 6 out of the first 36 clusters have strong model separation (15% of clusters). This allows the model, for example, to pull stereotypical representations that are more common to one of the model into its own cluster. Note that this is more prevalent in the 48-cluster setting. In the 12-cluster setting, only one cluster is dominated by a single model, corresponding to stereotypical depictions of Native Americans (cluster 9). You can explore the clusters further using this tool: https://hf.co/spaces/tti-bias/diffusion-face-clustering
>
> > Given that it appears that the clusters split by model, what is the rational for clustering them all together? Would it not make more sense to generate a fixed number of clusters per model in its own embedding space based on the Identities images from the same dataset, and cluster the professions images per model based on that?
>
> We considered the very trade-off that the reviewer mentions here between evaluating each model on its own generations and using mixed-model generations for the clustering. The first method is more self-contained, but will make results obtained across models less directly comparable since they're using different implicit notions of diversity. It can still be helpful to compare generation settings for a single model, but is less generally useful. The second method is susceptible to work worse when there is a strong difference between the clustering and test-time generation, but is more portable. We chose the second one based on these considerations.
>
> **[Manual validation of captions]**:
>
> > The authors could do this by both analyzing the accuracy of the captioning model on an existing fairness dataset labelled with these protected attributes as a baseline. [...] I would also suggest the authors manually check the captioning demographic accuracy on some small portion of the dataset themselves.
>
> We ran early experiments along the lines suggested by the reviewer here, by looking at the professions inferred by the VQA model for different identity generation prompts - however, it quickly became clear that properly evaluating these biases to gain the necessary insights would have constituted its own separate research project, which we would love to explore further in future work.
>
> **[Documentating tools and datasets]**:
>
> We appreciate the reminder and have taken the reviewer's advice by adding data cards to our dataset which we are actively expanding (see e.g.: https://hf.co/datasets/tti-bias/professions-v2). We have also created an [interactive summary page](https://huggingface.co/spaces/tti-bias/stable-bias) that provides an overview of the analysis, as well as all exploratory tools and insights that can be obtained from them.
>
> **[Clarifications]**:
>
> > adjectives
>
> We decided not to include the adjectives line of inquiry and have decided to pursue it in future work.
>
> > Can the authors please clarify that my understanding is correct: the Professions dataset is used in 4.1 and the identities dataset is only used to generate the embedding clusters in 4.2?
>
> 4.1 only uses professions, 4.2 uses both together
>
> > The authors could also consider using [1] to de-bias the existing captioning model.
>
> We appreciate the pointer and agree it would be interesting, but we have no access to this specific implementation.
>
> > Can the authors clarify the difference between the clustering.
>
> Clusters are better represented as "topic models" of social markers than as single terms, and have the same ability to capture higher order correlations.
>
> **[Overflowing table]**
>
> The overflowing table has been reformatted.

---

### Official Review · Reviewer_Uoyq · 2023-07-24
**Review for Stable Bias:Evaluating Societal Representations in Diffusion Models**

**Rating:** 8
**Confidence:** 5
**Correctness:** The dataset is constructed in a compr…
**Clarity:** This paper is very well-written.

**Strengths:**

1. This paper has great review of prior work on this topic, including recent literature on the fairness of machine learning, as well as ongoing discussions on the representativeness of generated outputs in TTI models.

2. The authors have proposed a computational framework towards measuring representative in diffusion models, and conducted various experiments on a novel dataset.

3. The limitation, as well as ethnical and social discussion, section of this paper is well-written with important future work directions highlighted.

**Additional Feedback:**

N/A

**Documentation:**

Yes

**Ethics:**

Not likely

**Limitations:**

Limitations are greatly addressed. One aspect I see is users of this tool might use this as an "ethics washing" tool and claim that this score can be used as a justification for how their model is "fair" and "diverse".

**Opportunities For Improvement:**

1. While conceptually it was easy to follow the authors' goal/measure, I think the paper could really be strengthened if the authors could explicitly state (in either words and/or equations) what *is being calculated/proxied*

2. In addition, while it is very interesting to measure the diversity using the clustering framework in the image embedding space over all combinations of gender and race/ethnicy, I was wondering whether it would be interesting to zoom in on a particular combination. In particular, understanding how much variation there is when we ask TTI to generate "white man" versus "black woman" might give us insights on something more direct? I would love to see extensions/discussions of this point further in this paper.

3. Some of the analysis choices are not spelled out: for instance, it says 3150 prompts in Figure 1, how are they generated (146 professional * 68 identity = 9K prompts?); how are 24 chosen (would the results be robust to a larger/smaller number)? Would the number of clusters affect what we are measuring here?

4. How do we interpret and benchmark some of the numbers generated? For instance, are users expected to refer to the numbers in a relatively way (e.g., hypothetically stable diffusion 1.4 has slightly better representative than stable diffusion 2.0), or would 0.5-0.8 (again just a goalpost number range) be generally a good/bad range for representativeness? Granted the numbers will be very model and dataset dependent, but I would love to see some more discussions here (spelling out my concerns in 1. might be of help?).

4. Much of the results is benchmarked against the US population/statistics -- do we expect that to be a good baseline? How do we see extending the experimental setup more internationally?

5. (minor) explain what markedness mean?

6. (minor) Could we try generating multiple race/gender in one pic for these tti systems and see what happens? e.g. a prompt such as "generate photo portrait of a panel of firepersons/fireflighters with diverse gender and race". While this does not quite answer the question of interest, I would love to see how and if that yields interesting results relevant to this work.

**Relation To Prior Work:**

Prior work has been greatly mentioned and attributed; I would love to see a bit more quantitative discussion/computation of these metrics, especially given the ever increasing volume of the generated images.

**Summary And Contributions:**

This paper proposed a systematic way to explore social biases in text-to-image systems, where outputs could be synthetic and non-human-like in nature. The approach relies on characterizing variations in the generated outputs by enumerating attributes of interest to diversity, such as gender and race, in the prompts. They investigated their methods empirically on three popular TTI systems (Dalle2, stable diffusion v1.4 and v2), and concluded that the existing systems tend to underrepresent marginalized identities to different extents.

---

> ### Author Response · Authors · 2023-08-18
>
> We are grateful to Reviewer Uoyq for their helpful suggestions.
>
> We address the main points that they raised below:
>
> **[What is being calculated]**:
>
> Our aim is to measure the gender and identity group differences for images generated by different text-to-image models for the same set of prompts. We do not, however, readily see how to capture the analyses using equations, as diversity and representativeness are multi-faceted concepts: our methodology supports various aspects in various ways. For example, in Table 3, the left and middle cells help quantify to what extent the models exacerbate existing social trends, while the right column focuses on the relationship between socio-economic status and gender and ethnic diversity.
>
> **[Honing in on specific examples]**:
>
> This is a great idea, and we will add an examination of how the variation varies when prompting "white man" versus "black woman".
>
> **[Providing more information about our clustering approach]**:
>
> We actually tried several cluster numbers (as can be seen in the [Cluster explorer tool](https://huggingface.co/spaces/tti-bias/cluster-explorer) that we created), but found that 24 was the optimal number of clusters in terms of distinctiveness and interpretability of the analysis. This is currently reported on in Appendix B where we evaluate the 12-cluster and 48-cluster settings, and show the findings to be fairly stable to the clustering size. We found that 24 provided the best trade-off between granularity and interpretability for our full list of social variables (3 marked gender phrases x 16 marked ethnicity phrases), and separated social variables without separating generation models as much. We will make sure to discuss that in detail in the final version of the paper.
>
> **[Expanding the discussion of results]**:
>
> We will use the extra page provided in the camera ready to discuss the significance of the results obtained and how to compare these numbers between different models and prompts, as well as how numbers obtained in different countries and regions (e.g. the EU) may differ, based on statistics from their respective institutions.
>
> **[Adding explanations]**:
>
> Thank you for pointing out our failure to explain the term *‘markedness’*- we had to remove it because we ran out of space in the paper. The short explanation ran as follows: “markedness is a linguistic concept that refers to the distinctiveness of a word or concept compared to others – in the case of professions, the fact that firefighters and pilots have distinctive uniforms that allow them to be easily distinguished from other professions.” We will add the short explanation back to Section 4.1 in the final version.
>
> **[Pursuing further experimentation]**:
>
> We really appreciate the reviewer’s idea regarding generating more images from diverse identity groups to study their distribution; we will pursue this analysis in our future work.

---

> > ### Comment · Reviewer_Uoyq · 2023-08-27
> >
> > Dear authors, thank you for your detailed reply. I confirm my initial assessment and remain extremely enthusiastic about this piece of work.

---

### Official Review · Reviewer_75aP · 2023-07-25
**Very interesting framework to audit Text-to-Image (TTI)  generative models, that could benefit from further details and documentation.**

**Rating:** 7
**Confidence:** 3

**Strengths:**

Nest I detail the main points I appreciate from the work:

- The problem under study is not only very interesting and timely, but also very difficult due to the lack of meta-information (on gender and ethnicity) on the artificially generated images.

- The three main components of the proposed auditing framework, in my humble opinion (as I do not have formal background in social science), are well motivated and executed for the problem under study.

- The authors apply two different text-marker tools of different nature to analyze the TTI models, obtaining coherent results between them. I believe this is key, as these methods have their own biases (as acknowledge by the authors in Section 5). Also, the The use of US labor demographics as a baseline, seems to provide a reasonable reference.

- The results, including those on the appendix, are very interesting and also worrying.

**Additional Feedback:**

I believe the potential impact of this work is large and thus, I encourage the authors to improve the presentation of their work (paper, appendix and  code/API) to clarify those points that remain unclear and provide also results on the interactive (visualization) part of their framework.

**Clarity:**

The paper is in general well written, but could benefit for further explanations on the methodology and results (see above). Yes, the authors did a good job, given the space constraints.

**Correctness:**

Everything looks correct to me, and both the data generation and the analysis  methods look reasonable  to me.

**Documentation:**

As mentioned above, in my opinion, there is significant room for documentation and details on the provided URL, which could nicely complement the details provided in the paper.

**Ethics:**

I do not detect major ethical problems but on the contrary, I hope this framework is a first step towards a better analysis of the bias of TTI (and other generative) models.

**Limitations:**

I do share the main limitations of the work discussed by the authors in Section 5. At the same time, I do still believe that the work is very interesting despite such limitations and may trigger significant future work on the bias auditing of generative models.

**Opportunities For Improvement:**

In general, i find the analysis very interesting and with a significant and also impactful contribution. However, I believe that the presentation of the approach and results could be further improved. Specifically, I detected the following main aspects for improvement:

- Description and caption of figures and tables could  be further improved. I find at times hard to understand and process the results presented in the paper. For example, it is unclear to me how to read the entropy column in Table 3 with respect to the BLS income value. Similarly, it is unclear to me what the "Ranks" in Figure 4 are and how are they computed.

- Regarding the clustering approach, I miss: i) further details on how the clustering method used; ii) results on  how changing the embedding extractor and clustering method would change the main insights.

- Missing results on the visualization tools described in section 4.3. Based just on the description of Section 4.3 , I find hard to assess how useful such qualitative analysis may indeed be.

- Appendix: it would be important to ensure the redability of the tables in the appendix, specially Table 18.

- URL documentation and navigation: I found hard to navigate the URL provided by the authors as almost no description is provided. That may as a consequence hinder the future use of the developed (and very nice) framework for auditing TTI models.

**Relation To Prior Work:**

Up to the best of my knowledge, the approach here is novel and I am not aware of similar published studies. The literature on the background section looks quite complete to me.

**Summary And Contributions:**

The paper focuses on the analysis of the sociology-demographic biases of data-driven Text-to-Image (TTI)  generative models. To that end, the authors run a comparative analysis between two different datasets generated using three  TTI models (Dall·E 2 , Stable Diffusion v 1.4 and 2): the "Identities dataset" that is generated based on combination of ethnicity and gender prompts, and the "Professions dataset" that aims to generate images of 146 professions without ethnicity and gender assignments. The analysis has three major components: i) based on the clustering of the embedding  of the generated images; ii) based on the text-feature extraction tool; and iii) based on visualization. The paper provides details and results mainly on the first two approaches, showing the quantitative differences in terms of the biases with respect to gender and ethnicity between the three studied models,  relying to US labor demographics as a references.  Finally, the authors describe the main limitations of their approach and future work.

---

> ### Author Response · Authors · 2023-08-18
> **Official Comment from Authors**
>
> We would like to thank Reviewer 75aP for their insightful suggestions and helpful feedback.
>
> Addressing the major themes that were raised:
>
> **[Improving the captions and the appendix]**:
> - The professions are ranked (sorted) by the relevant statistics provided by the US Bureau of Labor Statistics. So in Table 3, right, the "low 20" by rank corresponds to the 20 percent of job typed with the lowest median income in the US. In Figure 3, the jobs are grouped instead by the percentage of women (left), and the percentage of workers who self-reported their ethnicity as Black (right)
> - In Table 3, entropy provides a measure of diversity across a cluster. Professions with high entropy correspond to a mostly even distribution across clusters, and professions with low entropy are concentrated across a few clusters. Table 3 (right) shows a negative correlation between income and diversity across models, with the highest income jobs having the lowest entropy.
> - The upcoming revision will include more details to make the captions more informative and complete.
> - We have improved the presentation of the appendix and its readability by reformatting the overflowing table.
>
> **[Clustering approach]**:
> - further details on the clustering methods are provided in the (admittedly quite long appendix)
> - in particular, we direct the reviewer to tables 4 and 5 in the appendix, which compare the results obtained for different embedding extractors (ours and CLIP) and different numbers of clusters (12 and 48). Findings about the diversity of the different models' generations appear to be stable across extractors and numbers of cluster, we chose the BLIP VQA extractor because it appeared to separate social variables better (lower entropy in Table 4)
>
> **[Visualization tools]**:
> - the visualization tools provide a less structured and more free-form way of exploring model generations, and the ability for users to pursue narratives and categories that are of particular interest to them (e.g. clusters with a predominance of specific identity groups).
> - They also enable users to find visual characteristics regarding particular professions or identity groups based on color characteristics (e.g. that nurses typically wear blue scrubs, or Native Americans have colorful headdresses).
> - We have added
>
> **[Adding documentation to the tools and datasets created]**:
> - We have added a dataset card to the datasets that we have created as part of our analysis (e.g. https://hf.co/datasets/tti-bias/professions-v2) and are actively in the process of epxanding those.
> - We have also created an [interactive summary page](https://hf.co/spaces/tti-bias/stable-bias) to contextualize the various tools and to present observations that we have made using the tools.
>
>
> We hope that the changes above will be sufficient to improve the clarity and quality of our submission.

---

> > ### Comment · Reviewer_75aP · 2023-08-27
> > **Thanks and upgrade.**
> >
> > I would like to thank the authors for their responses. I have now decided to upgrade my score to accept.

---

### Author Response · Authors · 2023-08-31

We are grateful to the reviewers and their valuable comments. This has been a fruitful discussion which will undoubtedly help improve our paper.

Following some of the feedback, we have found it fitting to add three interactive explorers to complement the paper and its appendix:

- https://hf.co/spaces/tti-bias/stable-bias : A centralized listing of all interactive tools as well as a high level overview of analyses that our data and tools enable.
- https://hf.co/spaces/tti-bias/diffusion-clustering : A tool that enables the exploration of profession-level social biases over identity clusters.
- https://hf.co/spaces/tti-bias/diffusion-face-clustering : A tool that enables the image-based exploration of identity (e.g. gender and ethnicity) biases over the different clusters for different text-to-image models.

We have also added dataset cards to our datasets:

- https://hf.co/datasets/tti-bias/identities
- https://hf.co/datasets/tti-bias/professions
- https://hf.co/datasets/tti-bias/professions-v2

---

### Decision · Program_Chairs · 2023-09-22

**Decision:**

Accept (Spotlight)

**Comment:**

This paper presents a systematic approach to assess social biases in text-to-image systems, even when the generated outputs are synthetic and non-human-like. The study empirically evaluated this approach on three widely used TTI systems (Dalle2, Stable Diffusion v1.4, and v2) and found that these systems tend to underrepresent marginalized identities to varying degrees. The paper is recommended for an acceptance.